# Impacts of environmental conditions, and allelic variation of cytosolic glutamine synthetase on maize hybrid kernel production

Nardjis Amiour[1], Laurent Décousset[2], Jacques Rouster [2], Nicolas Quenard[2], Clément Buet[2], Pierre Dubreuil[2], Isabelle Quilleré[1], Lenaïg Brulé[1], Caroline Cukier[3], Sylvie Dinant [1], Christophe Sallaud[2], Frédéric Dubois[4], Anis M. Limami [3], Peter J. Lea[5] & Bertrand Hirel [1✉]

Cytosolic glutamine synthetase (GS1) is the enzyme mainly responsible of ammonium assimilation and reassimilation in maize leaves. The agronomic potential of GS1 in maize kernel production was investigated by examining the impact of an overexpression of the enzyme in the leaf cells. Transgenic hybrids exhibiting a three-fold increase in leaf GS activity were produced and characterized using plants grown in the field. Several independent hybrids overexpressing *Gln1-3*, a gene encoding cytosolic (GS1), in the leaf and bundle sheath mesophyll cells were grown over five years in different locations. On average, a 3.8% increase in kernel yield was obtained in the transgenic hybrids compared to controls. However, we observed that such an increase was simultaneously dependent upon both the environmental conditions and the transgenic event for a given field trial. Although variable from one environment to another, significant associations were also found between two GS1 genes (*Gln1-3 and Gln1-4*) polymorphic regions and kernel yield in different locations. We propose that the GS1 enzyme is a potential lead for producing high yielding maize hybrids using either genetic engineering or marker-assisted selection. However, for these hybrids, yield increases will be largely dependent upon the environmental conditions used to grow the plants.

[1] Institut Jean-Pierre Bourgin, Institut national de la Recherche Agronomique et de l'Environnement (INRAE), Centre de Versailles-Grignon, Unité Mixte de Recherche 1318, RD10, Versailles, Cedex, France. [2] BIOGEMMA-LIMAGRAIN. Site de la Garenne. Route d'Ennezat. CS 90126, Chappes, France. [3] Université d'Angers, Institut Agro, Institut National de la Recherche Agronomique et de l'Environnement (INRAE), Institute de Recherches en Horticulture et Smences (IRHS), Structure Fédérative de Recherche (SFR) Qualité et Santé du Végétal (QUASAV), Angers, France. [4] Ecologie et Dynamique des Systèmes Anthropisés (EDYSAN), Unité mixte de Recherche (UMR 7058), Centre National de la Recherche Scientifique (CNRS)-Université de Picardie Jules Verne (UPJV), Laboratoire d'Agroécologie, Ecophysiologie et Biologie intégrative, Université de Picardie Jules Verne, Amiens, Cedex, France. [5] Lancaster Environment Centre, Lancaster University, Lancaster, UK. ✉email: bertrand.hirel@inra.fr

Maize, (*Zea mays* L.), also called corn is now ranked first in terms of harvestable material production compared to all the other crops[1]. Recently there has been considerable interest in improving the nitrogen use efficiency (NUE) of such an important crop[2]. NUE is calculated as grain production per unit of N available. There are two primary components of NUE, which are referred to as 'uptake efficiency' (the efficiency of absorption or uptake of supplied N) and 'usage efficiency' (the efficiency with which the total plant N is used to produce grain)[3].

There have been an increasing number of studies to improve our knowledge of the limiting steps involved in the control of NUE in maize, in order to obtain maximal yield. The aim has been to identify which genes, physiology, biochemistry, and regulatory mechanisms are involved in the control of kernel production under limiting or non-limiting N fertilization conditions[4–6].

A limited number of genes involved for example in $NO_3^-$ uptake[7] and alanine biosynthesis[8], and glutamine synthesis[9,10] were proposed to be used either as breeding targets for engineering genetically modified plants with improved NUE and yield. Among these genes those encoding the cytosolic glutamine synthetase (GS1, E.C.6.3.1.2) isoenzymes appeared to be the most promising[11]. The reaction catalyzed by the enzyme GS, in conjunction with another enzyme called glutamate synthase (GOGAT), is considered to be the major if not the only route facilitating the incorporation of inorganic N into organic molecules.

For maize, the putative role of GS in kernel productivity was first suggested following a quantitative genetic approach, when QTLs for the leaf enzyme activity were shown to be coincident with QTLs for yield and its components. Later on, it was shown that total GS activity is an indicator that reflects the nutritional status of individual maize leaves with regards to N assimilation and recycling, whatever the level of N fertilization[12], thus strengthening the hypothesis that the GS enzyme plays a key role in plant productivity.

Functional validation of the two GS1 loci in maize was then undertaken using forward and reverse genetics. Interestingly, it was found that the GS1-3 isoenzyme is present in the leaf mesophyll cells (MCs) whereas the GS1-4 isoenzyme is specifically localized in the bundle sheath cells (BSCs) suggesting that they play a specific role in each cell type[13]. The impact of the knockout mutations *gln1-3* and *gln1-4* on kernel yield (total kernel mass per plant) and its components (kernel number per plant and thousand kernel weight) was studied in plants grown under controlled conditions. The phenotype of the two mutant lines was characterized by a reduction of thousand kernel weight in the *gln1-4* mutant and by a reduction of kernel yield in the *gln1-3* mutant. In the *gln1-3/1-4* double mutant, a cumulative effect of the two mutations was observed[13]. When grown under glasshouse conditions, transgenic maize lines overexpressing *Gln1-3* in the leaf mesophyll, exhibited an increase in kernel yield[13]. Field trials further confirmed that when *Gln1-3* or *Gln1-4* were overexpressed constitutively in maize lines, an increase in kernel yield accompanied by an enhancement of NUE was observed[14].

Despite the information available concerning the role of GS in maize lines, and the evidence of its importance in controlling kernel yield from the functional validation studies, it is important to know whether increasing the GS enzyme activity could also be beneficial at the level of test cross performance, which has, to our knowledge, never been conducted before. The objectives of the present study were thus: (1) to produce hybrids overexpressing cytosolic GS both in the mesophyll and in the bundle sheath, the two main maize leaf cell types in which a cytosolic form of the enzyme is the most active[13]: (2) to characterize the hybrids overexpressing GS1 encoded by the gene *Gln1-3* both at the biochemical and physiological level: (3) to perform replicated field trials at different sites in the USA exhibiting variable annual average temperatures, and during different years under various environmental conditions, such as mild water deficit: (4) to identify polymorphisms in the *Gln1-3* and *Gln1-4* candidate genes associated with kernel yield, using an association genetics study to select superior alleles that could be further used to breed maize hybrids with increased kernel production, as such a study has, to our knowledge, never been carried out.

## Results

**Production of maize hybrids overexpressing cytosolic GS.** To produce maize hybrids containing elevated amounts of GS1 activity in both the leaf MCs and BSCs, the *Gln1-3* full length cDNA, was fused to the cassava vein mosaic virus promoter (*CsVMV* promoter, allowing an overexpression of GS1 in the mesophyll cells) and to the promoter of the maize Rubisco small subunit (*RbcS*, directing overexpression of GS1 in the bundle sheath). *Gln1-3* cDNA was also used to direct its own expression in the BSCs because it was successfully overexpressed in transgenic maize lines[13]. Moreover, the *Gln1-3* protein sequence was 98% identical to that of *Gln1-4* with the same number of amino acid residues[15]. The two different promoter-*Gln1-3* gene fusions (plasmid pBIOS1458, Supplementary Fig. 1a) were then introduced into a recombinant plasmid (pBIOS1459) used for plant transformation *via Agrobacterium tumefaciens* (Supplementary Fig. 1b). Twelve independent transgenic hybrids (H12, H14, H17, H18, H20, H22, H23, H27, H31, H32, H39, and H40) containing up to two inserted T-DNA copies were selected for the different field trials. After an initial cross of the primary transformant ($T_0$ plant) with the pollen of the wild type (WT, A188), two rounds of self-pollination were performed in order to obtain plants for which the cob produced only homozygous seeds. Finally, in order to perform the field trials under agronomic conditions, homozygous *Gln1-3* overexpressing lines were crossed with a non-transgenic tester line (RBO1) to obtain hybrid seeds. In addition, *Gln1-3* overexpressing lines were crossed with two additional testers (AAX3 and AAX7) to examine the impact of GS1 overexpression on kernel yield in two other different genetic backgrounds. The number of independent transgenic hybrid lines used for the different field trials and for the characterization of the *Gln1-3* overexpressors was variable from one year to another depending on seed availability (Supplementary Data 1a). Control plants consisted of (1) a pool of two bulks of null segregants obtained from the cross between the tester line RBO1, (2) various homozygous plants derived from transgenic events for which the transgene was outcrossed during the first self-pollination, and (3) the wild type hybrid obtained from the cross of the WT with the tester line RBO1 (CH).

**Phenotypic characterization of maize hybrid lines overexpressing cytosolic GS.** Following a first field trial performed in 2010 (location Alleman, IA, USA: see the following Results section), for which an increase in kernel yield was observed in the transgenic hybrids overexpressing *Gln1-3* under the control of the *CsVMV* and *Rbcs* promoter, a second field experiment was conducted in order to perform a detailed phenotypic characterization of the transgenic hybrids. Plants were grown under optimal N feeding conditions (location Finch, MA, USA, in 2011 see Supplementary Data 1a, b). At the vegetative (V) stage, leaf GS activities of eight independent transgenic hybrids (selected from the twelve transgenic hybrids originally produced) were determined. The GS activities of plants overexpressing *Gln1-3* under the control of the *CsVMV* and *Rbcs* promoters (H12, H14, H18,

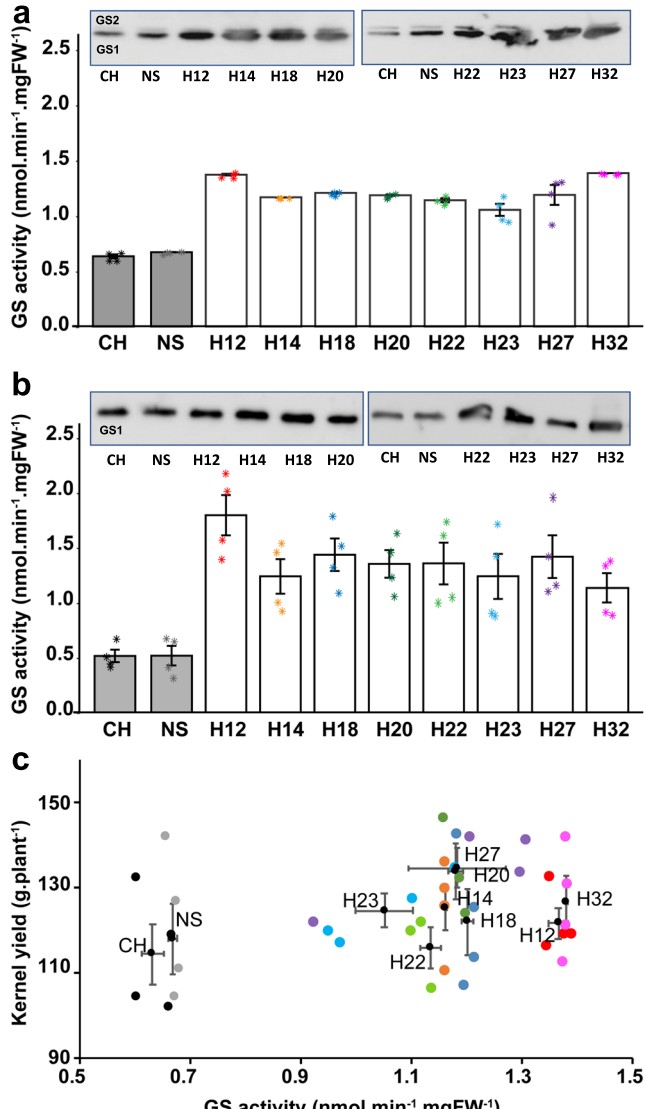

**Fig. 1 Characterization of the maize hybrids overexpressing *Gln1-3*.** The transgenic and control hybrids were grown in the field in 2011 in Finch (MA, USA). **a** Glutamine synthetase activity in leaves of untransformed control hybrids (CH), a pool of two bulks of hybrids null segregants (NS), and eight independent transgenic hybrids overexpressing *Gln1-3* under the control of the *CsVMV* and the maize *RbcS* promoters (H12 to H32) at the vegetative (V) stage. **b** Total leaf GS activity and subunit composition at the grain filling stage, 15 days after silking (15DAS). Total leaf GS activity was measured on four different plants for each of the eight independent transgenic hybrids and protein gel blot analysis was conducted using proteins extracted from a pool of the four different plants for each transgenic hybrid and controls. In panels **a**, **b** a protein gel blot showing the GS subunit composition in the leaves of the untransformed hybrids and null segregants and the eight hybrids overexpressing *Gln1-3* is shown. At the V stage, the upper band (molecular mass of 44 kDa) corresponds to the plastidic GS (GS2) subunit, and the lower band (molecular mass of 40 kDa) corresponds to the cytosolic GS (GS1) subunit for the vegetative (V) stage. At 15DAS only GS1 subunit was detected. The two separate sets of protein gel blots shown in both (**a**, **b**) were selected from the original gel blots presented in Supplementary Fig. 2. **c** Scatter plot showing the relationship between kernel mass per plant and total leaf GS activity at the V stage in untransformed control plants and the eight independent hybrids overexpressing *Gln1-3*. Kernel mass per plant (g) was measured on four plants for each transgenic hybrid grown in the field in 2011 (location Finch, MA, USA) under non-limiting N fertilization conditions. Values are the mean ± SD for GS activity and yield. For the eight transgenic hybrids, the yield increase was on average 7% higher compared to the CH and the NS ($P = 0.04$). Symbols asterisks (**a**, **b**) and dots (**c**) for the values of the four plants: CH: black, NS: gray, H12: red, H14: orange, H18: blue, H20: dark green, H22: light green, H23: light blue, H27: purple, H32: pink.

H20, H22, H23, H27, and H32), were similar and approximately twice as high as that detected in the untransformed control hybrid (CH) and the bulk of null segregants (NS) (Fig. 1a and see Supplementary Data 8). An increase in the total leaf GS activities was also observed in the transgenic hybrids at 15 days after silking (15 DAS), being approximately three-fold higher compared to the CH and the NS. As for the vegetative stage, the increase in total leaf GS activity at 15DAS was comparable in the eight transgenic hybrid lines (Fig. 1b and see Supplementary Data 8).

The leaf GS isoenzyme protein content of the CH, the NS, and the eight GS overexpressors was examined using protein gel blot analysis. At the vegetative stage, in the leaves of the CH, two polypeptides were detected. The most abundant 40 kDa polypeptide corresponded to the cytosolic form of GS (GS1), whereas the 44 kDa polypeptide, which was barely detectable, corresponded to the plastidic form of GS (GS2). Compared to the CH and the NS, a three-to four-fold increase in the amount of GS1 protein was observed in the eight GS overexpressors, (Fig. 1a). When the leaf GS proteins were analyzed at 15DAS, a two-fold increase in GS1 protein content was determined in the transgenic hybrids. At this later stage of plant development, GS2 protein in the leaves was not detected (Fig. 1b).

Originals of the protein gel blots used to show the GS subunit composition in the controls and the GS overexpressors are shown

in Supplementary Fig. 2. An immunocytochemical study was employed to determine in which leaf tissue GS was expressed in the transgenic hybrids. At the vegetative stage of plant development, the GS proteins were localized in leaf sections of the CH and the hybrid H27 overexpressing *Gln1-3* using immunogold labeling (Fig. 2). In the CH leaf tissue sections treated with gold-labeled tobacco anti-GS antibodies, GS was detected both in the mesophyll cells (MCs), and the bundle sheath cells (BSCs), the fluorescent signal being much higher in the latter (Fig. 2b). In leaf sections of the GS hybrid line H27, the fluorescent signal was much stronger in both the mesophyll cells and BSCs compared to the untransformed control hybrid. A clear signal was also observed in the cytosol of the companion cells present in the phloem tissue (cc) (Fig. 2b). In a control leaf section treated with a pre-immune serum both the mesophyll cells and the BSCs were unstained (Fig. 2c). To improve the resolution of the immunolocalization technique, immunogold TEM experiments were carried out on the same leaf blade sections. Quantification of gold particles in the CH and in H27 showed that GS was located both in the cytosol and the plastids of the mesophyll cells and of the bundle sheath cells (Supplementary Table 1). The amount of plastidic GS protein was similar in the mesophyll cells and in the BSCs, while the amount of GS protein in the cytosol in the mesophyll cells was lower than that present in the BSCs. In the control and transgenic hybrid H27, similar amounts of GS protein were detected in the plastids. In the transgenic hybrid, a three-fold increase in the amount of cytosolic GS was observed in both the mesophyll cells and in the BSCs. A two-fold increase in GS protein was also observed in the phloem companion cells present in the vasculature of the BSCs. Quantification of the gold particles confirmed that there was a correspondence with the three-fold increase in total leaf GS activity (Fig. 1a and see Supplementary

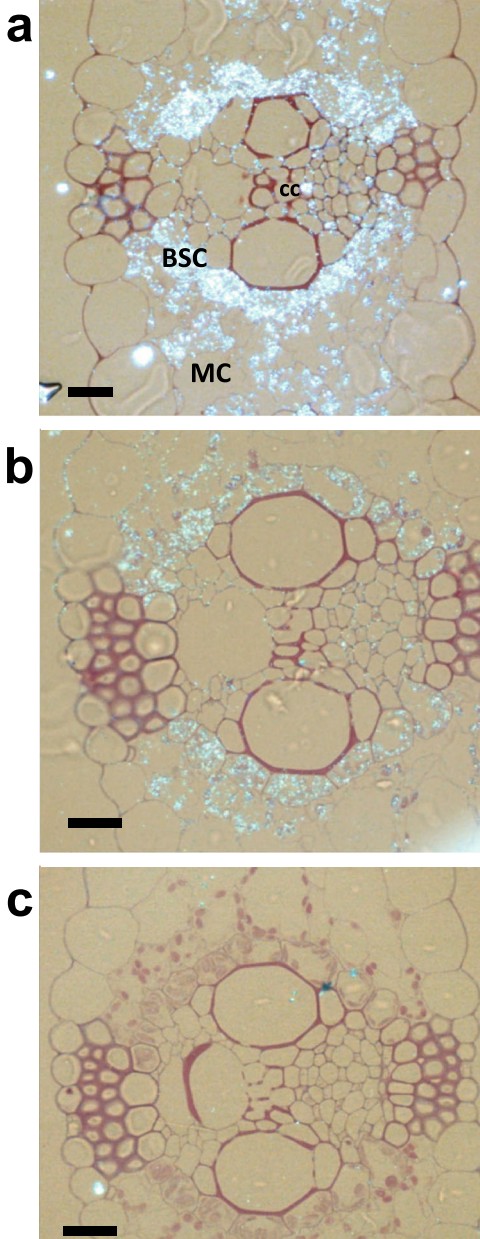

**Fig. 2 Histological immunolocalization of GS in transverse leaf sections of a transgenic maize hybrid overexpressing *Gln1-3* under the control of the *CsVMV* and maize *RbcS* promoters.** Sections were examined under bright-field epipolarized light. **a** Immunolocalization of GS in the leaf of a transgenic hybrid overexpressing *Gln1-3* under the control of the *CsVMV* and of the maize *RbcS* promoters (hybrid line H27). **b** Immunolocalization of GS in untransformed hybrids. **c** Control section treated with preimmune serum. *BSC* bundle sheath cell; *cc* phloem companion cells; *MC* mesophyll cell. Bars = 100 μm.

Data 8), also indicating that such an increase was similar in the mesophyll cells and BSCs.

The impact of GS overexpression on plant phenotype and kernel production was determined using the eight independent transgenic hybrids grown in the field in 2011 (location Finch, MA, USA). At plant maturity, no significant differences were seen in the dry matter accumulation of the vegetative parts of the shoots. In contrast, a significant increase in kernel yield, expressed as mass per plant (g/plant) was observed, which when compared to the CH and the NS was on average 7% higher

($P = 0.04$), (Fig. 1c, see Supplementary Data 2 and 8 for the statistical analysis). Although the increase in leaf GS activity was similar in the eight transgenic hybrids, this increase was not necessarily correlated with the increase in kernel yield. For example, although the increase in leaf GS activity was similar in H20 and H22, the increase in kernel yield was 15% higher in the former. In H12 and H32, the increased kernel yield was the average of that obtained with the eight hybrids overexpressing GS1, even if their GS activity was approximately 15% higher compared to the other hybrids. No differences between the hybrids overexpressing GS1 and the controls (CH and NS) were observed for the leaf C and N concentrations at 15DAS and at maturity. In contrast, there was a low but significant increase ($P = 0.03$) in the total N content of the kernels of the GS overexpressors, which was on average 9% higher (Supplementary Fig. 3). Details of the measured phenotypic traits and the statistical analyses are presented in Supplementary Data 2.

A $^{15}NH_4^+$ labeling experiment was conducted in order to determine whether increased GS activity in the leaves also modified the dynamics of amino acid accumulation. Detached leaves of untransformed control (CH) and three independent transgenic hybrids (H12, H22, and H32) that exhibited an increase in kernel yield in the field were fed for 8 h with a nutrient solution containing 4 mM $^{15}NH_4^+$. In the three transgenic hybrids, $^{15}NH_4^+$ was incorporated into the main amino acids present in the leaves, notably into the two N amide atoms of glutamine and asparagine (Supplementary Data 3). The highest $^{15}$N-labeling was detected in alanine, the amino acid that predominated in the soluble pool of all the lines. In the three transgenic hybrids, there was a two-fold increase in both glutamine content and $^{15}NH_4^+$ incorporation in the glutamine pool.

**Field trials of transgenic maize hybrids overexpressing *Gln1–3*.** In the first stage of field assessment, an experiment was conducted in 2010 (location Alleman, Al, USA) using the control hybrids (CH), a bulk of null segregants (NS), and ten independent transgenic hybrids (H12 to H32) cultivated in a random complete block design. Compared to the two types of control hybrids (CH and NS), the increase in kernel yield, expressed as quintals per ha (Ql. ha$^{-1}$) in the hybrids overexpressing *Gln1-3* was an average of 5.2% ($P = 0.041$), (Fig. 3a site Al indicated by a gray arrow and see Supplementary Data 9). Details of the data and of the statistical analyses for this first field trial are presented in Supplementary Data 4, (sheet: Yield Alleman 2010). In contrast, no significant differences were obtained for thousand kernel weight, (Supplementary Data 4, sheet: TKW Alleman 2010).

Additional field trials were then conducted by replicating trials further at nine additional locations in the USA, in four different years in 2011, 2013, 2014, and 2015.

The different transgenic and control hybrids used in the field trials are presented in Supplementary Data 1, together with the field trials performed in different locations in the USA characterized by variable annual average temperatures. Details of the plot design, number of transgenic events, number of replicates for each transgenic event, and data for kernel yield and kernel moisture are presented in Supplementary Data 4 (sheet: Yield raw data all years) and summarized in Fig. 3a (see Supplementary Data 9). Data and details of the statistical analyses are presented in Supplementary Data 4: sheet Statistics locations. It can be seen that when all the transgenic events were compared to the controls in the different locations, a significant increase in kernel yield (from 5 to 11%, $p < 0.05$) was obtained in four sites out of 14, the highest increase being obtained in 2013 in the locations George (Ge) and Ashkun (As) and the lowest at Aleman (Al) in 2010 and at Sleepy Eyes (SE) in 2015 (see Supplementary

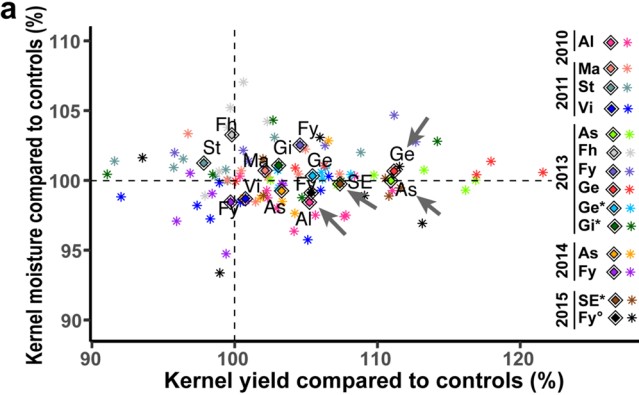

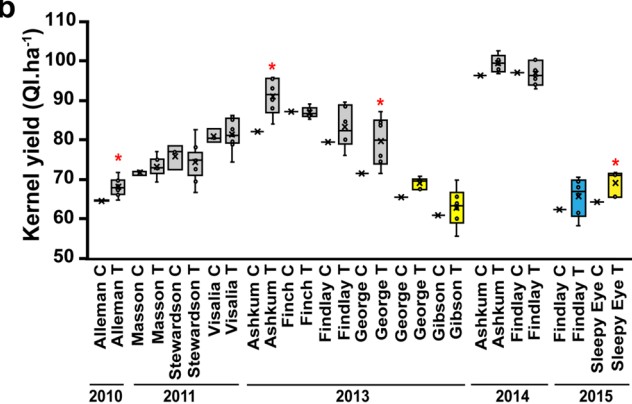

**Fig. 3 Impact of *Gln1-3* overexpression on kernel yield of maize hybrids grown in the field over five years in different locations. a** Relationship between kernel yield (measured in Ql. ha$^{-1}$) and of kernel moisture (% compared to the control hybrids and null segregants) corresponding to the difference between the mean of all the different transgenic hybrids overexpressing *Gln1-3* and the mean of the two types of control hybrids (colored diamonds). The colored asterisks represent the different transgenic events tested in the different locations over the five years of experimentation: Alleman (Al) in 2010 (pink). Masson (Ma), (orange); Stewardson (St) (green) and Visalia (Vi) (dark blue) in 2011. Ashkum (As), (light green); Finch (Fh); (light gray); Findlay (Fy), (violet); George (Ge), (red); George (Ge*),(cyan) and Gibson (Gi*), (dark green) in 2013. Ashkum (As), (yellow−orange) and Findlay (Fy), (dark purple) in 2014. Sleepy Eyes (SE*), (brown−orange) and Findlay (Fy°) (black) in 2015. Presence of a moderate water deficit in George (Ge*) and Gibson (Gi*) in 2013, Sleepy Eyes (SE*) in 2015, and water-saturated soil conditions in Findlay (Fy°) in 2015. The data are from Supplemental Data 4 in which the number of transgenic events, the years of experimentation, the locations, the experimental design, and the statistical analyses are presented. The list of different transgenic events used in the field experiments is summarized in Supplementary Data 1a. The significant increase in kernel yield in a location (*p* < 0.05) is indicated by a gray arrow Alleman (Al) in 2010, Askhum (As) in 2013, George (Ge) in 2013, and Sleepy Eyes (SE) in 2015. **b** Box plots showing details of kernel yield obtained over the five years of experimentation (2010, 2011, 2013, 2014, and 2015) in the different locations in the control hybrids (C) and in the transgenic hybrids (T). The yellow boxes indicate the presence of a moderate water deficit and the blue symbol water-saturated soil conditions. The black dots in the box represent the different transgenic events and the horizontal line in the box the average value for kernel yield. The Significant increase in kernel yield in a location (*p* < 0.05) is indicated by a red asterisk.

Data 1b for the different locations in the USA). In SE in 2015, a moderate water deficit occurred at flowering. In the four sites for which kernel yield was increased in the hybrids overexpressing *Gln1-3*, the variation in grain moisture was on average less than 2% and not significantly different between the transgenic and the control hybrids (*P* > 0.05), (Supplementary Data 4, sheet: Statistics locations).

In order to determine if a significant difference could be observed, a two-way statistical analysis was then performed using a mixed model restricted maximum likelihood (REML) grouping the 14 field experiments. This was performed to measure differences in yield between the two types of controls (CH and NS) and the transgenic hybrids overexpressing *Gln1-3*. A test of the differences in terms of the least square means was made for the fixed effects between the control hybrids (CH and NS) and all the transgenic hybrids (Supplementary Data 4, sheet: ANOVA yield all sites). Compared to the different control hybrid lines, a 3.8% increase in kernel yield was obtained (*P* = 0.029) in the different hybrid lines overexpressing *Gln1-3*. In addition, we found that the impact of the interaction between the different transgenic events and the different environments (year, location, and water stress) was not significant (*P* = 0.74).

Details of the changes in kernel yield between the different locations and over the five years of experimentation are presented in Fig. 3b (see Supplementary Data 9). It can be seen that kernel production was variable from one year to another, as well as from one location to another in the same year. When a significant increase in kernel yield was observed in the *Gln1-3* over-expressors, it occurred whether kernel yield was low (e.g., around 67 Ql. ha$^{-1}$ in Alleman in 2010) or high (e.g., around 90 Ql. ha$^{-1}$ in Ashkum in 2013). In some locations such as Findlay in 2013, the average increase in kernel yield was not significant. This was due to the fact that only 2 (H14 and H32) out of the six independent transgenic hybrids tested had a higher kernel weight per plant compared to the control hybrids (Supplementary Data 4, sheet Yield per Event Location Year, Gray highlight).

The impact of *Gln1-3* overexpression on kernel yield was further assessed using multi-environmental analysis. To achieve this the four transgenic events H14, H20, H22, and H32 were selected as they were those that had been the most frequently tested across several years and in different locations, (Supplementary Data 1a, b). Compared to the different control hybrids, a 3.9% increase in kernel yield was obtained (*P* = 0.05) in the different hybrid overexpressing *Gln1-3*. The impact of the interaction between the four transgenic events and the different environments was not significant (*P* = 0.85), (Supplementary Data 4, sheet Statistics multi-local analysis). In the field trial performed in 2015 in Sleepy Eye with the five transgenic hybrids produced with the tester RBO1 (H14, H20, H22, H32, and H39), we examined if the increase in kernel yield was also observed when *Gln1-3* overexpressing lines were crossed with two other different testers 5 (AAX3 and AAX7). On average, compared to the control hybrids, a 7.4% increase in kernel yield was obtained (*P* = 0.01) with the three types of hybrids (Supplementary Data 4, sheet: Tester interaction). In addition, we observed that there was no significant interaction between the five transgenic events and the three different testers used to produce the hybrids (*P* = 0.98).

**Association analysis in *Gln1.3* and *Gln1.4* with kernel yield.** A panel of 375 Double Haploid Inbred Lines (DHIL) derived from a Multiparental Advanced Generation Inter-Cross (MAGIC) maize population was used to test associations between SNPs in *Gln1-3* and *Gln1-4* gene sequences, and kernel yield. In the *Gln1-3* gene,

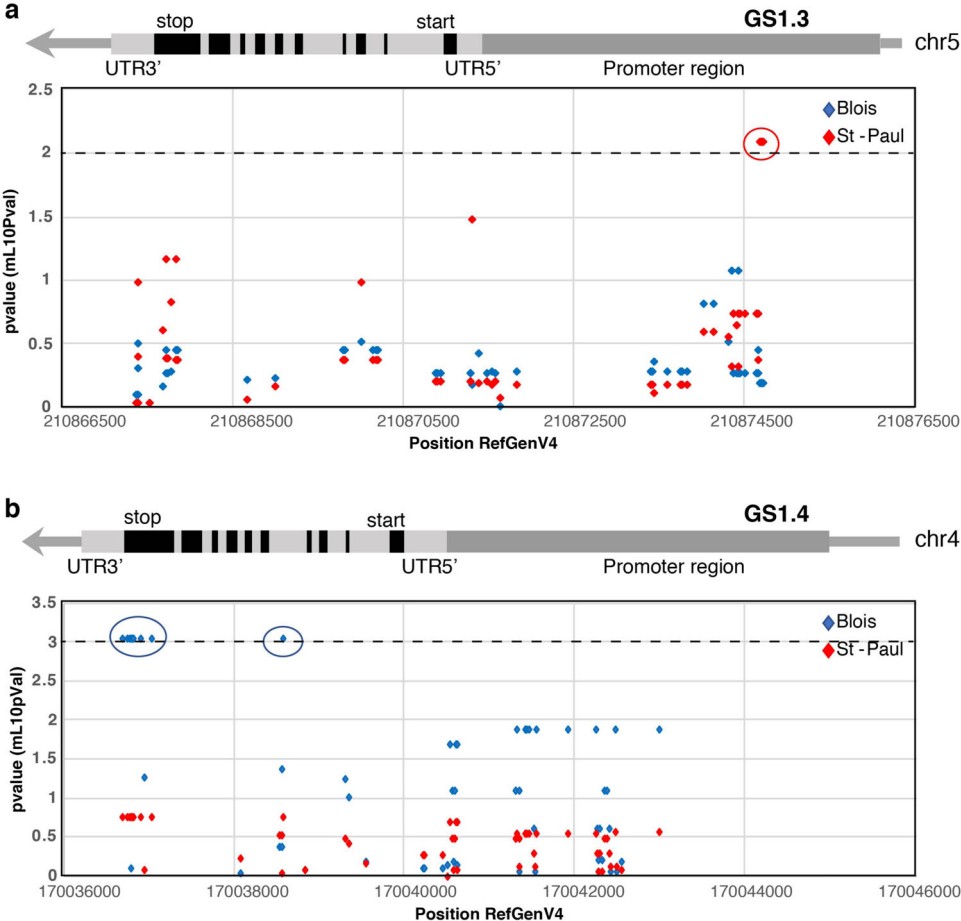

**Fig. 4 Manhattan plots showing SNP association with the genes encoding *Gln1-3* and *Gln1-4*.** Manhattan plots showing association mapping *p*-values associated with kernel yield at 15% moisture (KY15) and genomic location of SNPs located in gene encoding *GS1.3* (**a**) and *GS1.4* (**b**). Each symbol represents the *p*-value of a single SNP associated with a trait measured in the field trial conducted in Blois in 2015 (blue) and in Saint-Paul in 2016 (red). The horizontal dashed line represents the significant threshold with a −log₁₀(*p*-value) >2 and >3 for *Gln1-3* and *Gln1-4* respectively. The six TAS markers (at markers C05M60, C05M61, C05M62, C05M63, C05M64, and C05M65) for *Gln1-3* and the nine TAS markers (at markers C04M01, C04M02, C04M03, C04M05, C04M06, C04M07, C04M08, C04M10, and C04M15) for *Gln1-4* are indicated by circles. In the upper part of the figure is indicated the position of the introns, exons, 3′ UTR, 5′ UTR, and promoter region of the two genes.

**Table 1 SNP association with the genes encoding *Gln1-3* and *Gln1-4*.**

| Year | Location | *p*-value *Gln1-3* | *p*-value *Gln1-4* |
|------|----------|-------------------|-------------------|
| 2015 | Blois | <2 | >3[a] |
| 2015 | Saint-Paul | <2 | >2[a] |
| 2016 | Saint-Paul | >2[a] | <2 |
| 2017 | Blois | >2[a] | <2 |
| 2017 | Nerac | >2[a] | <2 |

[a]Significant threshold with a −log₁₀(*p*-value) >2 and >3 for *Gln1-3* and *Gln1-4*

six Trait Associated Single nucleotide polymorphism (TAS) markers (at markers C05M60, C05M61, C05M62, C05M63, C05M64, and C05M65) were detected in the gene promoter region (from 3509 to 3535 bp from the start codon). For example, in the field trial which was performed in the location Saint-Paul in 2016, TAS marker association with kernel yield adjusted at 15% moisture content (KY15) was found to be associated at a significant level of 1% (*p*-value = 0.008) as illustrated in Fig. 4a (see Supplementary Data 10). Similar results were obtained in three other environments (Saint-Paul in 2015, Blois and Nerac in 2017) as summarized in Table 1. Parameters of the SNP

association study in the different locations and over the different years are presented in Supplementary Data 5. Pairwise Linkage Disequilibrium (LD) measured among the SNP in the *Gln1-3* genomic region revealed that the six identified TAS were in complete LD ($R^2 = 1$). These six TAS were located between 210.874.701 and 210.874.727 bp in the promoter region of *Gln1-3* (maize B73 reference genome RefGen_v4, Jiao et al., 2017). LD between the TAS markers and the markers nearby was never higher than 0.43 (Supplementary Fig. 4a and Supplementary Data 6). The minor allele frequency (MAF) for all the TAS markers was 0.17 on 352 DHILs of the panel and the allele substitution effect (ie. the difference between the genotypic values of the two genotypes AA and BB at a given marker) was 3.82 Ql.ha⁻¹, for a KY15 mean value of 93.0 Ql.ha⁻¹. In the other location (Blois in 2015) no significant TAS was detected in the *Gln1-3* gene region.

For *Gln1-4* gene region nine TAS markers (markers C04M01, C04M02, C04M03, C04M05, C04M06, C04M07, C04M08, C04M10, and C04M15) were found to be associated with KY15 at a significant level of 0.1% (*p*-value = 0.0008) in the location Blois in 2015 as illustrated in Fig. 4b (see Supplementary Data 10). Similar results were obtained in another environment (Saint-Paul in 2015) as summarized in Table 1. The first marker was positioned 10 bp after the stop codon in the 3′UTR region.

The following seven markers were detected in the exon10 of the gene while the last one was found in the 4th intron. Parameters of the SNP association study are presented in Supplementary Data 5. The nine identified TAS were in complete LD ($r^2 = 1$) and located in the *Gln1-4* 3′UTR, exon10, and intron4 regions, between position 170.036.686 bp and 170.038.578 bp of the maize B73 genome (RefGen_v4; Jiao et al., 2017). Between the TAS markers and the markers nearby, LD was never higher than 0.36($r^2$), (Supplementary Fig. 4b and Supplementary Data 6). The MAF for the TAS markers was 0.06 on 344 DHILs of the MAGIC panel and the allele substitution effect was 5.54 Ql. ha$^{-1}$, for a KY15 mean value of 84.6 Ql. ha$^{-1}$. In the second location (Saint Paul in 2016) no significant TAS was detected in the *Gln1-4* gene region.

## Discussion

In this study, we have further assessed in a more robust manner the functional importance of cytosolic GS (GS1) in the control of kernel production in maize. Transgenic hybrids overexpressing the enzyme in the leaves, were produced and tested under agronomic conditions over several years and in different locations in the USA. Such a large-scale field testing has, to our knowledge, never been conducted on maize hybrids to perform a functional validation of an enzyme involved in N metabolism. Moreover, maize represents an essential dual-use food and energy crop[1,16], which could lead to use GS1 as a breeding target to increase kernel yield in commercial hybrids.

Homozygous transgenic maize lines overexpressing *Gln1-3*, a gene encoding cytosolic GS, were first produced and then crossed with a non-transgenic inbred tester line (used to produce commercial elite varieties), to obtain transgenic hybrids overexpressing the GS1 enzyme both in the mesophyll cells and bundle sheath cells , two different cell types in which the native GS1 is present[13]. An increase of at least three-fold of total leaf GS1 enzyme activity was obtained in the transgenic hybrids. In the leaves, the dual cell-specific overexpression of GS1 protein was confirmed using both light and transmission microscope immunological studies. When $^{15}NH_4^+$ was provided to detached leaves of the hybrids overexpressing GS1, a two-fold increase in the rate of glutamine synthesis was observed over a short period of labeling. Such results indicate that the genetically modified plants have the capacity to assimilate more inorganic N. This inorganic N not only drives kernel set but is also used for kernel filling[17,18].

The three-fold increase in total leaf GS activity in the hybrids overexpressing the gene *Gln1-3* encoding GS1 was almost identical in several independent transgenic events. Such findings mean that the tester lines used to produce hybrids do not have any major impact on the expression of the transgene, as is the case for a number of native genes that do not exhibit a midparental level of expression[19]. Such results open up the possibility of producing commercial hybrids exhibiting enhanced GS1 activity in a reproducible manner, which prevents the variability generally observed in the level of expression of transgenes due to random insertion into the genome[20,21]. Such random insertion of the transgene could explain why the increase in leaf GS1 activity was not necessarily correlated with the increase in kernel yield in some of the transgenic hybrids. Nevertheless, the range of variation observed between the two traits did not exceed 20%, indicating that such variability had no major impact on kernel production.

When a series of 14 field experiments, conducted over 5 years in ten different locations were analyzed as a whole, on average a 3.8% increase in kernel yield was observed in the GS1 overexpressors. Interestingly, in one field trial performed in 2015, we also observed that in two other types of hybrids produced by crossing GS1 overexpressing lines with two other different testers, an average increase of 7.4% in kernel yield could be obtained. Such results, strongly suggest that the positive impact of GS1 overexpression on kernel production is independent of the genetic background. At a glance, such an increase seems to be relatively modest. However, maize production is a very large and significant market with the export value reaching 33.9 billion dollars in 2018 (https://www.mccormick.it/as/all-the-latest-data-on-maize-production-around-the-world/). Thus, a 3−4% increase in maize yield would potentially provide substantial additional incomes to both the farmers and to the economy of many countries in the world. However, detailed analyses of these field experiments revealed that a significant increase in kernel yield (from 5 to 11%, $p < 0.05$) was only obtained in four sites out of 14 in 2010, 2013 and in 2015. Such results indicate that both the year and the site of experimentation had a major impact on the productivity of the transgenic hybrids. Therefore, multienvironmental analysis of the yield data, was further conducted to evaluate the combined impact of the location and of the year of experimentation on kernel production. To achieve this, four independent transgenic hybrids that were present in the majority of the field trials were selected. We observed that the year of experimentation had a marked impact on plant performance, since a significant increase in kernel yield was observed in two locations out of five in 2013. In addition, we observed that out of the four transgenic hybrids, only two of them (H14 and H20) exhibited a significant increase in kernel yield in four different locations and that their locations were not necessarily the same for each hybrid. These results show that the environmental conditions and the choice of the transgenic event are two important components that need to be simultaneously considered in order to optimize the impact of GS1 overexpression on plant productivity. However, it seems that the most significant increases in yield were obtained in the central USA in which the largest number of field trials were conducted. This observation suggests that the GS overexpressors could be more adapted to temperate climatic conditions. However, an increase in kernel yield was also obtained in a western warmer region, indicating that more trials need to be performed under such climatic conditions. This finding is in line with a recent study in which maize phenotypic variance, heritability, and yield relationships were evaluated across different environments, leading to the conclusion that it will be difficult to find a case where a single gene improves a single trait that leads to consistent yield[22].

To strengthen the genetic engineering approach, breeding for a higher yield in maize will require the identification of superior alleles for *Gln1-3* and *Gln 1-4*. Genome-wide association mapping is a powerful method for identifying alleles affecting a variety of quantitative agronomic traits in many crops, including maize[23,24]. Up to now, in crops, associations were only found between grain size traits of bread wheat and polymorphic regions of one gene encoding GS1[25]. In durum wheat allelic variants of plastidic GS (GS2) were also associated with grain protein content[26] and SNP loci were associated with cytosolic GS (GS1) and grain quality[27]. In contrast, no information is available on the genetic diversity of the maize genes encoding GS and their allelic effect on yield-related traits, notably in hybrids that are used as commercial products. Thus, in order to identify polymorphisms in the *Gln1-3* and *Gln1-4* candidate genes associated with kernel yield, we have applied an association study in a MAGIC-derived panel (Supplementary Fig. 5). SNP variation for these two genes in maize was analyzed and associations between *Gln1-3* and *Gln1-4* and KY were searched. Although low, significant associations were found for six TAS in the promoter region of *Gln1-3*. Higher significant associations were found for *Gln1-4* for nine TAS located in the 3′ untranslated region, in the tenth exon, and in the

fourth intron. For the two genes encoding GS1, all these significant associations were found with KY. In other genes such as *VRN-1* controlling vernalization both in barley and wheat, such types of polymorphism have already been found beyond coding regions for agronomic traits related to yield[28]. In the present investigation, both in *Gln1-3* and *Gln1-4* genes, the favorable alleles were significantly associated with an increase in KY of 4.1 and 6.5% respectively, compared to that obtained with hybrids carrying the unfavorable alleles. Such an increase in KY in the range of 5% was comparable to what was obtained in the transgenic hybrids overexpressing *Gln1-3* both in the mesophyll and bundle sheath leaf cells (3.8%), two cells types in which GS1 is the most active. This provides an indication on the expected increase in KY under agronomic conditions if GS1 is to be used as a breeding target using marker-assisted selection. However, for the gene encoding *Gln1-3*, associations with KY for *Gln1-3* were only significant in three environments out of five over two consecutive years, whereas for *Gln1-4* there were significant in two different locations only in 2015 (Table 1). Again, this finding indicates that environmental conditions must be considered in order to optimize the use of GS1 as a breeding target to increase plant productivity. Plant phenotypic adaptation to the environment is modulated by a myriad of interacting genes and metabolic pathways, including epigenetic processes[29]. It can therefore be hypothesized that under different environments and by virtue of their tissue-specific location, the level of GS1 activity in the mesophyll and the bundle sheath cells of transgenic or conventionally bred plants will be independently or simultaneously controlled leading to an increase in kernel yield only when the environmental conditions are the most favorable.

In conclusion, the results obtained in the present study, together with those reported in previous quantitative genetic[30] and functional validation studies, further confirm that cytosolic GS can significantly contribute to maize kernel production. Therefore, we propose that the GS1 enzyme is a potential lead for producing high-yielding maize hybrids. However, to obtain a significant and stable increase in kernel yield over time, selection of the best GS1 overexpressors will be required. As proposed by Xu[29], further studies will be required for phenotype prediction of maize plants modified or selected for higher GS activity integrating genotype, type of environment, and plant developmental stage interactions.

## Methods

**Plant transformation, regeneration, and characterization**. Maize transformation of the inbred line A188 with the *Agrobacterium tumefaciens* strain LBA4404 harboring a super-binary plasmid was performed according to the protocol of Ishida et al.[31]. In particular, the composition of all media cited hereafter is detailed in the reference above. The protocol was slightly modified concerning the selective marker, which was the *NPTII* gene instead of the *bar* gene. The super-binary plasmid used for transformation was the result of a recombination within the *Agrobacterium tumefaciens* strain between plasmid pBIOS 1459 and plasmid pSB1 (harboring the *virB* and *virG* genes isolated from pTIBo542[32]). Plasmid pBIOS1459 contains between the T-DNA borders, a neomycin resistance cassette (*NPTII* gene[33]) flanked by an *Oryza sativa* (Os) actin promoter with its first intron and 3' *Nos* terminator). This selectable marker cassette was flanked by 2 DS elements (5' Ac and 3'Ac[34]). Two copies of a synthetic *Gln1-3* full length cDNA[15] (accession numbers D14577.1 and X65928) including their 3' non-coding regions, one flanked by the cassava vein mosaic virus promoter (CsVMV promoter[35]) fused to the rice actin1 first intron[36] to enhance the activity of CsVMV promoter and the 3'*Nos* terminator and the other one flanked by the promoter of the maize Rubisco small subunit (*RbcS*)[37] and the 3'Nos terminator. The nucleotide sequence of *Gln1-3* was 98% similar at the amino acid level to that of *Gln1-4* the other gene encoding GS1[13]. The resulting recombinant plasmid used for transformation was pBIOS1459 (Supplementary Fig. 1). Transgenic plants were then cultivated in a glasshouse (18–24 °C) from April to September 2008 and either selfed or pollinated with the WT line A188 to produce seeds. For each transgenic line, the number of T-DNA copies was determined by q-PCR using a primer amplifying a T-DNA specific fragment located between the selection cassette and the cassette with the gene of interest (Forward: CCGTCCCGCAAGTTAAATATGA and Reverse:

GCTTAGATCTGAGATCGGTAAGGAA), (See Supplementary Fig. 1, for the position of the primers). Twelve independent transgenic hybrid lines (H12, H14, H17, H18, H20, H22, H23, H27, H31, H32, H39, and H40) containing up to two inserted T-DNA copies were selected for the different field trials. After an initial cross of the primary transformant (T$_0$ plant) with pollen of the wild type (WT) A188, 2 rounds of self-pollination were performed in order to obtain plants for which the cob carried only homozygous seeds. Finally, in order to perform the field trials under agronomic conditions, homozygous *Gln1-3* over-expressing lines were crossed with a non-transgenic tester line (RBO1) to obtain hybrid seeds. In addition, transgenic hybrids were produced by crossing the line overexpressing GS1 with two other different testers AAX3 and AAX7. The RBO1 line belongs to the Stiff Stalk heterotic group, whereas AAX3 and AAX7 belong to the OH43 and Iodent*OH43 heterotic groups respectively.

**Plant material for agronomic and physiological studies**. Twelve maize (*Zea mays*, L.) hybrids in which the plasmid pBIOS1459 was introduced (H12, H14, H17, H18, H20, H22, H23, H27, H31, H32, H39, and H40), two bulks of hybrid null segregants (named T01581-BULK-NS and T01594-BULK-NS and a control untransformed wild type hybrid (CH) were grown in the field. The two bulks of hybrid null segregants were obtained from the cross between the tester line RBO1 and various homozygous plants derived from transgenic events for which the transgene was outcrossed during the first self-pollination. Untransformed control hybrids corresponding to the cross between A188 and RBO, were used either as the female or male parental lines. In addition, maize hybrids (H14, H20, H22, H32, and H39) were produced, using two other testers (AAX3 and AAX7). Transgenic and control hybrids were grown at different locations in the USA: Alleman (IA, N 41.834911/W -93.656433), in 2010, Visala (CA, N 36.539310/W -121.242780), Stewardson (IL, N 39.383640/W -88.552070), Finch (MA, N 41.968333/W -93.605097) and Mason City (IA, N 43.331338/W -93.088322) in 2011, Gibson (IL, N 38.659080/W -121.751742), Ashkum (IL), Findlay (OH, N 39.521080/W -88.775610), Finch (MA, N 41.968333/W -93.605097), George (WA, N 38.712022/W -121.763260) in 2013, Findlay (OH, N 39.521080/W -88.775610) in 2014, Sleepy Eye (MN, N 44.369494/W -94.675001) and Findlay in 2015. The locations and their yearly average temperature are shown in Supplementary Data 1b. The list of the hybrids tested over different years for either their biochemical characterization or for their agronomic performance is presented in Supplementary Data 1a. The number of independent transgenic hybrid lines used for the different field trials was varied from one year to the other depending on seed availability. The level of N fertilization was 175 kg/ha supplied before sowing and N provided by the soil was estimated at 60 kg/ha. Both phosphorus (P$_{205}$) and potassium (K$_{20}$) were also applied at 100 kg/ha. Water deficit experiments were conducted in order to attain a decrease in kernel yield of approximately 20–30%. The intensity of the drought stress was controlled by monitoring soil moisture using a watermark granular matrix sensor[38].

The control and transgenic hybrids were grown in a random block design, completely balanced lattice or split plot depending on the year of experimentation and on the location, with four to five replicates for each hybrid line (Supplementary Data 4). An outside border area of at least six rows (commercial variety 356M70 from Blue River Organic Seed, IA, USA) was planted surrounding the different plots of the field trial. Each plot was composed of two rows of maize of 6 m length. The number of plants in each plot was between 60 and 75, depending on the location.

The seeds were sown in May in the five years of experimentation. Kernel yield components at plant maturity were measured using a plot combine harvester (Juniper Systems, Inc., Logan, UT, USA). Kernel yield (KY), thousand kernel weight (TKW), and kernel moisture (KM) were measured using the on-board equipment of the plot combine harvester. Kernel weight was then normalized to moisture at 15% using the following formula: Normalized kernel weight = measured KW × 100-measured moisture (expressed in %)/85 (100-normalized soil moisture at 15%). In all the field experiments kernel yield was expressed in quintal per hectare (Ql. ha$^{-1}$).

For all biochemical analyses performed at the vegetative stage (V) and at 15DAS leaf and kernel samples were harvested in the year 2011 in Finch (MA, USA) using a pool of the two BULKS-NS (NS) and the wild type hybrid (CH) as controls, and the transgenic lines H12, H14, H18, H20, H22, H23, H27, and H32 (Supplementary Data 1a). As 25% of the plants were removed from the different plots or used for leaf sampling, this 2011 field trial which was partly destructive was not included in the evaluation of yield over the five years of experimentation. At the vegetative (V) stage, half of the 6th fully emerged leaf without the main central midrib was harvested at the 7–8 leaf stage between 9 a.m. and noon. For the grain filling stage, half of the leaf below the ear was harvested 15 days after silking (15DAS), each plant being harvested at the same developmental stage. The plant developmental stage at 15DAS has been shown to provide a good indication of the transition occurring when both C and N metabolites start to be actively translocated to the developing kernels[6]. Moreover, the leaf below the ear was selected, since it provides a good indication of the sink to source transition during grain filling[39]. No major delays in silking dates were observed between the different hybrids. The leaf samples were immediately placed in liquid N$_2$ and stored at −80 °C until further analysis. In the same experiment, in order to evaluate the agronomic performance of the *Gln1-3* overexpressors in comparison to the controls, shoot DW 15DAS,

shoot DW at maturity, and kernel mass per plant (g/plant) were measured. The leaf C and N contents were also measured on plants harvested 15DAS and at maturity (kernel harvesting) and in the kernels.

**Metabolite extraction and biochemical analyses**. Leaf GS activity was measured using the biosynthetic reaction using hydroxylamine instead of ammonium as a substrate leading to the formation of γ-glutamyl hydroxamate as previously described[13]. The total C and N content of 25 mg of frozen leaf material and dry kernels was determined in an elemental analyzer using the combustion method of Dumas (Flash 2000, Thermo Scientific, Cergy-Pontoise, France). For nitrate and ammonium measurements, 100 mg of frozen leaf powder were extracted in 1 mL of 80% ethanol at room temperature for an hour. During extraction, the samples were continuously agitated and centrifuged at $12,000 \times g$ for 5 min. The supernatants were removed and the pellets were subjected to a further extraction in 60% ethanol and finally in the water. All the supernatants were combined to form the water/ethanol extract. Nitrate and ammonium were determined by the method of Cataldo et al.[40] and by the phenol hypochlorite assay[41].

**$^{15}$N-labeling experiment**. The $^{15}NH_4^+$ labeling experiment was conducted using detached young developing leaves collected at the vegetative (V) stage of plant growth and development. The atom %$^{15}$N of each amino acid was then determined by GC-MS analysis according to the protocol described by Cukier et al.[42]. In this protocol, the methods are described for tracing the pathway by which plants are able to take up $^{15}$N-labeled ammonium and convert them into amino acids. Following amino acid extraction, purification, and separation by GC/MS, a calculation of the $^{15}$N enrichment of each amino acid is carried out on a relative basis to identify any differences in the dynamics of amino acids accumulation. Plants were grown under hydroponic conditions on a complete nutrient solution[30] and the 6th emerged leaves from four individual maize plants for the control and transgenic hybrid lines were used to perform a 6 h labeling experiment with 4 mM $NH_4Cl$, enriched with 50% $^{15}NH_4Cl$.

**Protein gel blot analysis**. Soluble proteins were extracted from frozen leaf powder that had been previously stored at −80 °C. Extraction was performed at 4 °C in a buffer containing 50 mM Tris-HCl pH 7.5, 1 mM EDTA, 1 mM $MgCl_2$, 0.5% (w/v) polyvinylpyrrolidone, 0.1% (v/v) 2-mercaptoethanol and 4 mM leupeptin, and separated by SDS-PAGE[43]. The percentage of polyacrylamide in the gels was 10% and equal amounts of protein (10 μg) were loaded onto each track. Proteins were electrophoretically transferred to nitrocellulose membranes for protein gel blot analysis. GS1 and GS2 polypeptides were detected using polyclonal antisera raised against GS2 of tobacco (*Nicotiana tabaccum* L.) diluted 10 times using the protocol described by Hirel et al.[44]. Soluble protein was determined using a commercially available kit (Coomassie Protein assay reagent, Biorad, München, Germany) using bovine serum albumin as a standard.

**Immunolocalization studies**. For light microscope immunological studies, leaf pieces (2–3 mm[2]) were fixed in freshly prepared 1.5% (w/v) paraformaldehyde in phosphate buffer 0.1 M, pH 7.4 for 4 h at 4 °C. For immunolocalization, material was dehydrated in an ethanol series (final concentration 90% v/v ethanol) and embedded in London Resin white resin (Polysciences, Warrington, PA). Polymerization was carried out in gelatin capsules for 10 h at 54 °C. For light microscope immunological studies, thin sections of 1 μm were floated on drops of sterile water on slides and treated at room temperature in 1% periodic acid for 1 h, followed by 0.5 M $NH_4Cl$ for 1 h. Sections were then washed three times with distilled water and incubated for 1 h at room temperature in T1 buffer (0.05 M Tris-HCl buffer containing 2.5% (w/v) NaCl, 0.1% (w/v) BSA, and 0.05% (v/v) Tween 20, pH 7.4). Sections were first incubated with 10% BSA. Sections were first incubated with 5% normal goat serum in T1 buffer for 1 h at room temperature and then with anti-GS rabbit serum diluted 1:70 in T1 buffer for 6 h at room temperature. Sections were washed five times with T1 buffer, twice with T2 buffer (0.02 M Tris-HCI buffer containing 2% NaCl, 0.1% BSA, and 0.05% Tween 20, pH 8) and incubated with 10 μl colloidal gold-goat anti-rabbit immunoglobulin complex (Sigma, St. Louis, MO, USA) diluted 1:50 in T2 buffer for 2 h at room temperature. Immuno-gold labeling was revealed by silver enhancement as described by the supplier (British Biocell International) and sections were back-stained with 1% fuchsine before microscopical examination under bright field and/or epipolarised light on a Nikon Eclips 800 epifluorescent photomicroscope. Controls were run either by omitting the primary antibody or by its substitution with preimmune serum. For immunolocalization examination by transmission electron microscopy (TEM) immunological studies, leaf pieces (2 to 3 mm[2]) were fixed in freshly prepared 1.5% (w/v) paraformaldehyde in phosphate buffer 0.1 M, pH 7.4 for 4 h at 4 °C. Leaf material was dehydrated in an ethanol series (final concentration 90% v/v ethanol) and embedded in London Resin white resin (Polysciences, Warrington, PA). Polymerization was carried out in gelatin capsules for 10 h at 54 °C. For immuno-transmission electron microscopy studies, ultra-thin sections were mounted on 400 μm mesh nickel grids and allowed to dry at 37 °C. Sections were first incubated with 5% (v/v) normal goat serum in T1 buffer for 1 h at room temperature and then for an additional 6 h at room temperature with the GS antiserum, also used to perform protein gel blots, diluted 100 times in T1 buffer. Sections were then

washed three times with T1 buffer and incubated for 2 h at room temperature with 10 nm colloidal gold goat anti-rabbit immunoglobulin complex (Sigma, St Louis, Mi) diluted 70 times in T1 buffer. After several washes, grids were treated with 5% (w/v) uranyl acetate in water and examined with a Philips CM12 electron microscope (Philips, Eindhoven, The Netherlands) at 100 kV. Negative controls were conducted by substituting the serum-containing GS antibodies with pre-immune rabbit serum.

**Production of the MAGIC panel and genotyping**. The maize MAGIC panel used to perform association studies, was created from a funnel cross of 16 founder lines (B96, EP1, DK105, FV2, CO255, F492, A654, FV252, DK63, C103, OH43, A632, B73, W117, ND245, and VA85) representing the most significant heterotic groups used for maize hybrid production in temperate regions (similar to the panel produced by Dell'Acqua et al.[45]). The original population called BALANCE was used to extract 375 DHIL at the 3rd generation of mixing (G8). The crossing scheme used to create the MAGIC panel is illustrated in Supplementary Fig. 5. The 375 DHILs along with the 16 founder lines and the tester were genotyped with the Axiom maize 600 K array[46]. The parental origin of each allele in the DHILs was inferred using the R/qtl package[47] and each DHIL was represented as a mosaic of the founder genomes. Knowing the parental mosaics of each DH lines, the SNP detected from the sequencing of the founders were projected on each DHIL (allowing *in silico* genotyping). The MAGIC panel allowed to detect more than 12 million single nucleotide polymorphisms (SNPs) following next generation sequencing (NGS) data of the 16 founders and of the tester line. Among these 12 million SNPs, approximately 8 million were selected using different criteria of quality control recommended by the US HapMap[48] including identification by descent (IBD)[49] and linkage disequilibrium (LD). These selected markers were positioned on the maize reference genome B73 RefGen_v4[50]. The SNPs located in the coding sequences of the genes *Gln1.3* (65 SNPs) and *Gln1.4* (56 SNPs), in the promoter (5′) and 3′ terminal region, were used to perform the association genetics studies. Thus, a given Filtered Gene Set (FGS), (gene coordinate RefGen_v4)[50] was extended with 5000 bp into the putative promoter region[51] and extended with 500 bp into the 3′ terminal region, for the final selection of SNPs.

**Association study**. The association genetics study was performed using SNP markers identified by sequencing the founder lines of the MAGIC panel, targeting two candidate genes *Gln1-3* and *Gln1-4*, including their promoter and 3′ terminal regions as described above. The SNP markers were anchored on the maize genome RefGen_v4[50].

**Field trials and phenotyping of the MAGIC panel**. The DHILs of the panel were crossed to the tester line MBS847. The test-cross progenies were evaluated in the field for yield, in two different locations and over two years of experimentation. The experiments were designed to limit confounding effects due to differences in precocity, notably by reducing the range of flowering time. Kernel yield was adjusted to 15% moisture. Two field trials were conducted in Blois (France, 47.74683°N 1.23296°E, altitude = 123 m) and in Saint-Paul-lès-Romans, France, 45.041260°N 5.070226°E, altitude = 185 m) in 2015 using 346 hybrids. A third field trial was conducted in Saint-Paul-lès-Romans, 45.07464°N 5.11341°E, altitude = 185 m) in 2016 using 380 hybrids. Two additional field trials were conducted in Blois (47.444498°N 1.135907°E, altitude = 120 m) and in Nerac (France, 44.101156°N 0.18223°E, altitude 55 m) in 2017. The total level of N fertilization was 175 kg/ha (two successive applications of 150 kg/ha and 200 kg/ha in the form of urea at 1 leaf and 8 Leaves stages respectively and N provided by the soil was estimated at 60 kg/ha. Both phosphorus ($P_{205}$) and potassium ($K_{20}$) were also applied at 100 kg/ha. The experiments were carried out following an alpha-lattice design with two replicates for each trial. The plot length was comprised between 5.35 and 5.2 m with two rows spaced by a 0.8 m interval in the different locations. Temperature, rainfall, solar radiation, and water potential at 30, 60 et 90 cm depths were recorded each day to monitor the trials and especially the irrigation status to avoid a water stress to occur. The two first trials were sown on April 23rd 2015 in Blois at a density of 95000 seeds/ha in a clay-sand soil and 86250 seeds/ha in Saint Paul in a sandy silty clay soil. The third trial was sown on May 6th 2016 in Saint Paul at a density of 90000 seeds/ha. In 2017 the two trials were conducted in Blois and in Nerac at a density of 90,000 and 85,000 seeds/ha respectively in a silty clay soil for the later. In Saint Paul in 2015 a thermal stress defined as hot temperature scenario[52] occurred at flowering and during the grain filling period whereas in Nerac in 2017 it was only during the later. Two applications of N were scattered over the plots at 1 Leaf and 8 Leaves stages. Weeds, diseases, and pests were controlled using conventional agronomic practices for both trials. The adjusted mean values for kernel yield are presented in Supplementary Data 7.

**Statistics and reproducibility**. For the combined agronomic and physiological studies performed on plants grown in the field in Finch in 2011, the control and transgenic hybrids were grown in a random block design with four blocks. For all biochemical analyses, shoot, and kernel mass production in each of the four different blocks, six leaf samples or plants were harvested each coming from a different plant. The six leaf samples or plants were pooled, making four replicates in

total. All data shown in the graphs are presented as the mean with S.D. One-way ANOVA statistical analyses were performed using a Student−Newman−Keuls test (t-test) to identify differences between the controls and the transgenic hybrids ($p \leq 0.05$). For the protein gel blot analysis, the four soluble protein extracts used to measure total leaf GS activity were pooled, each containing 2.5 μg of proteins. For the $^{15}NH_4+$ labeling experiment, four individual maize plants for the control and transgenic hybrid lines were used to perform the labeling experiment. Statistical analysis was performed by a t-test, which was used to identify amino acids exhibiting a different pattern of $^{15}N$-labeling ($p \leq 0.05$). For light microscope immunological studies, gold particles were counted on 10–15 different leaf sections. Significant differences at $p \leq 0.05$ were identified using the t-test. For all the other field experiments performed to measure differences in yield between the controls and the transgenic, a two-way statistical analysis was performed using a mixed model restricted maximum likelihood. The model is the following: random factors = replicate (complete), sub-block (ibloc) tested in the replicate (for lattice only). Fixed factor (construct). A test of the differences in terms of the least square means was made for the fixed effects at the construct level between the control hybrids (CH and NS) and the transgenic hybrids. The association between the SNPs, and kernel yield measured in different years and locations, was performed using a mixed linear model (MLM) implemented through ASReml statistical package that fits linear mixed models using residual maximum likelihood (REML)[53], (VSN International Ltd) on R environment[54]. The model used was the K model[55]. The association study was conducted using the leave one chromosome out (LOCO) method in which the kinship is calculated from all the SNPs but those located on the chromosome where the tested marker is located[56,57]. For the field trials and phenotyping of the MAGIC panel, the experiments were carried out following an alpha-lattice design with two replicates for each trial. The adjusted mean values for kernel yield were calculated.

**Reporting summary**. Further information on research design is available in the Nature Research Reporting Summary linked to this article.

## Data availability

This data set and all other source data generated or analyzed during this study are included in this published article and its supplementary information files (Supplementary Data 1–10). A full scan of protein gel blot is shown in Supplementary Fig. 2. Additional details can be obtained from the corresponding author on reasonable request. The plasmid PBIOS1458 described in this paper (Supplementary Fig. 1) is available at Addgene under the number 171715. The other recombinant Ti-plasmid used for transformation pBIOS1459 is not available as it was used by Biogemma-Limagrain under a license of Japan Tobacco International. The genotypic information for the association genetics study is available at the EVA website (https://www.ebi.ac.uk/eva/?Home) under the reference ERZ3038796 for the analyses and PRJEB46968 for the project.

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

## Acknowledgements

We thank C.F. for preparing the constructs; C.J., A.B., A.G., and C.S. for maize transformation; S. L. and E.B. for taking care of the transgenic plants in the greenhouse; I.M., A.L., and A.S. Canoy for molecular analyses; I.D. for the field trial applications in the USA and B.K., D.E., C.A., and K.A. for nurseries management and field trials with the transgenic plants. We also thank Alain Murigneux and Morgan Renault for their contribution to the creation of the Maize MAGIC Population, Marie-Hélène Tixier and Jeremy Lopez for the management of the field trials for the association genetics study, Magalie Leveugle, Jean-Philippe Pichon, and Hervé Duborjal for their involvement in the genotyping by sequencing project; Johan Robin, Nicolas Sounac and Frédéric Sapet for their involvement in in silico genotyping. We are grateful to Sébastien Praud as research team supervisor for the association study. The study received the financial support of Genoplante (project GNP_B04) and of ANR/Genoplante (project NUE-MAIZE PCS-09-GENM-127).

## Author contributions

N.A., I.Q., and L.B. performed the phenotypic characterization of the transgenic plants; L.D., C.B., and P.D. performed the association genetic studies; N.Q. conducted the field trials; C.K. and A.L. performed the 15N-labeling studies; J.R., and C.S. designed the constructs, conducted the maize transformation experiments including the production of transgenic hybrids; F.D. performed the immunolocalization studies, B.H., P.J.L., L.D., and S.D. wrote the paper and prepared the figures and the data files; B.H., C.S, and P.D. designed and supervised the different experiments.

## Competing interests

The authors declare no competing interests.
