## [Peer Review File · Communications Biology]

Reviewers' Comments:

Reviewer #1:

Remarks to the Author:

The study of Amieur et al., describes the impact of glutamine synthetase (GS) in maize kernel size. The authors performed overexpression of GS by generating transgenic lines and most importantly exposed these lines in field trial experiments with different environmental conditions over five years. Additionally, polymorphisms in the genetic loci of GSs were associated with the kernel yield - further supporting that GS expression can directly affect the kernel size.

The manuscript is descriptive, the experiments well performed and the rationale easy to follow. I have just a few minor comments that, in my opinion, would improve the manuscript:

In Figure 1, the line H23 seems to be more expressed than line H32 (Western blot in panel 1a) but the GS activity seems to be higher in H32 although the kernel yield is almost identical (panel 1c). Similarly, both the GS activity and the GS expression in lines H22 and H20 are almost identical (panels 1a, 1b) but the kernel yield is apparently higher in H20 (panel 1c). Could maybe the authors include a few lines (either in the results or in the discussion section) providing possible explanations for this outcome?

Line 441: The accession number cited "d14577x65928" does not give any result in Genbank. Is there a typo possibly? Also in lines 440 & 445, a synthetic cDNA designed based on codon optimization is mentioned. I encourage the authors to provide the final synthetic sequence used (either as a supplemental file or as an accession number from Genbank) since it may be helpful to others performing similar research.

Extended data Fig. 1. Please explain briefly (either in the legend or in the corresponding subsection of the Methods) why the 1st intron of OsActin is included after the CsVMV promoter and why two copies of the 3'Nos terminator is used in the construct. This information may not be that obvious to all the readers of this study.

Line 454: Extended Data Fig. 1 is cited in this line but I do not see the mentioned qPCR.

Reviewer #2:

Remarks to the Author:

The authors present results of a multi-year evaluation of a transgenic glutamine synthetase trial together with an association mapping effort to determine if native allelic variation at GS loci in maize contribute to differences in grain yield. At a high level, the reporting of a transgenic technology which can increase grain yield on average by 3-4% averaged over a number of events and environments is an important finding. Taken together with evidence for native allelic variation association with grain yield of the GS loci studied, the authors conclude that GS is a potential lead for producing high yielding maize hybrids. In this respect, I believe the authors are correct in stating that GS is an important and promising target for improvement. The authors also acknowledge the role of trait x environment interaction and recognize that further optimization of the top GS1 overexpressors will be required. I agree with this conclusion as well.

The manuscript will be of interest to the broader field of crop scientists both as regards the importance of GS as a target for improving nitrogen use efficiency/grain yield and as a demonstration of a single gene manipulation to achieve significant intrinsic yield improvement through a strategic metabolic pathway. The authors are to be commended for the persistent efforts over multiple years to compile the data sets which led to these conclusions.

I believe that the work is convincing and sufficient to support the conclusions, even though it could

certainly be augmented in important ways. The most important limitation of the transgenic trials is the limitation to one genetic background. It appears that all evaluations were conducted by crossing the A188 transformation line to one tester. The strategy to evaluate multiple events against both the wild-type and negative segregant controls gives a robust (and quite promising) result in this specific background, however the significant question with physiological traits which remains unanswered in this experiment is the degree to which genotype x trait interaction exists (i.e. how penetrant is the transgene in other genetic backgrounds or how stable is the effect over various hybrid backgrounds). Given the timeframe of the experiments it seems plausible that the authors could have tested the A188 material on multiple testers and even possibly introgressed one or two lead events into elite lines for hybrid evaluation. Presenting data on additional testers and or in entirely different genetic backgrounds (as would be necessary to determine how efficacious the transgene would be across a commercial lineup of products) would definitely strengthen the manuscript.

That said, I feel that the authors are somewhat understated in the remarkable impact that this group of GS events demonstrates. The win percentage rate of figure 1a is really very promising, and the impact of GS overexpression in the four most widely studied transgenic events was nearly 4%. This represents 2 to 3 years of breeding progress for grain yield. It seems plausible from the data presented that a prioritized GS1 overexpressor could be identified which would consistently provide yield advantage in a majority of environments with minimal potential risk of lower yield. I believe that this level of improvement in performance would be appreciated by farmers as varietal replacement of hybrids in the market place is routinely occurring based on smaller incremental improvements. I do appreciate that the authors acknowledge the existence and importance of the trait x environment interaction they report, but I feel that they perhaps overstate the implication since this exists when comparing any two entities in any wide-scale field trial evaluations.

The primary limitation of the association mapping study is that the experiment was only conducted in two locations. This is very limited for the question that the researchers are trying to address.

That said, the MAGIC population was well designed and the testcross approach appropriate. The finding that both Gln1.3 and Gln1.4 native alleles associated strongly with yield in one of two locations contributes to the evidence base that GS1 variation is important. Although this would have been strengthened by additional environments, the authors conclusions are appropriately made. The results suggest an important role of GS1. Taken together with the transgenic results, this does point to a plausible target for further validation work for either a marker-assisted breeding or gene editing approach.

I do think that the results of this study will influence thinking in the field. There is general skepticism about the potential for manipulation of specific genes to increase grain yield, and documenting a ~4% yield improvement by overexpression of one gene is an important piece of evidence for those working in this space to consider.

I do feel that the statistical approaches used are appropriate. I also appreciate the level of detail provided by the authors given the multi-year, large scale effort of this project. Although it would be very difficult to exactly replicate the work, I feel that it is described in adequate detail to give knowledgeable practioners a very good idea of how the authors carried out the research.

Respectfully, Michael (Mike) Olsen, CIMMYT Global Maize Program

Reviewer #3:
None

Reply to reviewers: Manuscript COMMSBIO-20-2908-T

Reviewers' comments:

Reviewer #1

(Remarks to the Author):

The study of Amiour et al., describes the impact of glutamine synthetase (GS) in maize kernel size. The authors performed overexpression of GS by generating transgenic lines and most importantly exposed these lines in field trial experiments with different environmental conditions over five years. Additionally, polymorphisms in the genetic loci of GSs were associated with the kernel yield - further supporting that GS expression can directly affect the kernel size.

The manuscript is descriptive, the experiments well performed and the rationale easy to follow. I have just a few minor comments that, in my opinion, would improve the manuscript:

In Figure 1, the line H23 seems to be more expressed than line H32 (Western blot in panel 1a) but the GS activity seems to be higher in H32 although the kernel yield is almost identical (panel 1c). Similarly, both the GS activity and the GS expression in lines H22 and H20 are almost identical (panels 1a, 1b) but the kernel yield is apparently higher in H20 (panel 1c). Could maybe the authors include a few lines (either in the results or in the discussion section) providing possible explanations for this outcome?

We have included a few sentences in the revised version of the manuscript page 9 in the Results section and page 15-16 in the Discussion to present and discuss the point raised by reviewer 1.

Line 441: The accession number cited "d14577x65928" does not give any result in Genbank. Is there a typo possibly? Also in lines 440 & 445, a synthetic cDNA designed based on codon optimization is mentioned. I encourage the authors to provide the final synthetic sequence used (either as a supplemental file or as an accession number from Genbank) since it may be helpful to others performing similar research.

We have corrected the presentation of the two accession numbers on page 20 in the Materials and Methods section so that can be used to find GS1 cDNA sequences in the Genbank database.

D14577.1

LOCUS MZEGS1B 1350 bp mRNA linear PLN 08-JAN-2003

DEFINITION Zea mays mRNA for glutamine synthetase, complete cds.

ACCESSION D14577

X65928

LOCUS X65928 1317 bp mRNA linear PLN 18-APR-2005

DEFINITION Z.mays mRNA gs1-3 for glutamine synthetase.

ACCESSION X65928

Extended data Fig. 1. Please explain briefly (either in the legend or in the corresponding subsection of the Methods) why the 1st intron of OsActin is included after the CsVMV promoter and why two copies of the 3'Nos terminator is used in the construct. This information may not be that obvious to all the readers of this study.

We have modified the text both in the Materials and Methods section in page 20 and in the legend of Extended Data Figure S1 to make it clear to the reader.

Line 454: Extended Data Fig. 1 is cited in this line but I do not see the mentioned qPCR.

We have indicated the position of the two primers in Extended Data Figure1 and modified the text page 20.

Reviewer #2 (Remarks to the Author):

The authors present results of a multi-year evaluation of a transgenic glutamine synthetase trial together with an association mapping effort to determine if native allelic variation at GS loci in maize contribute to differences in grain yield. At a high level, the reporting of a transgenic technology which can increase grain yield on average by 3-4% averaged over a number of events and environments is an important finding. Taken together with evidence for native allelic variation association with grain yield of the GS loci studied, the authors conclude that GS is a potential lead for producing high yielding maize hybrids. In this respect, I believe the authors are correct in stating that GS is an important and promising target for improvement. The authors also acknowledge the role of trait x environment interaction and recognize that further optimization of the top GS1 overexpressors will be required. I agree with this conclusion as well.

The manuscript will be of interest to the broader field of crop scientists both as regards the importance of GS as a target for improving nitrogen use efficiency/grain yield and as a demonstration of a single gene manipulation to achieve significant intrinsic yield improvement through a strategic metabolic pathway. The authors are to be commended for the persistent efforts over multiple years to compile the data sets which led to these conclusions.

I believe that the work is convincing and sufficient to support the conclusions,

even though it could certainly be augmented in important ways. The most important limitation of the transgenic trials is the limitation to one genetic background. It appears that all evaluations were conducted by crossing the A188 transformation line to one tester. The strategy to evaluate multiple events against both the wild-type and negative segregant controls gives a robust (and quite promising) result in this specific background, however the significant question with physiological traits which remains unanswered in this experiment is the degree to which genotype x trait interaction exists (i.e. how penetrant is the transgene in other genetic backgrounds or how stable is the effect over various hybrid backgrounds). Given the timeframe of the experiments it seems plausible that the authors could have tested the A188 material on multiple testers and even possibly introgressed one or two lead events into elite lines for hybrid evaluation. Presenting data on additional testers and or in entirely different genetic backgrounds (as would be necessary to determine how efficacious the transgene would be across a commercial lineup of products) would definitely strengthen the manuscript.

Data have been added using two additional testers in the experiment performed in the location Sleepy eye in 2015.

These data are presented in Extended Data Set 4 R1, sheet Tester Interaction with additional information in Extended Data set 1 on the transgenic lines and controls used to produce hybrids with the two additional testers.

Results are presented on page 6 and 12 and discussed on page 16.

The Materials and Methods section has been modified accordingly.

That said, I feel that the authors are somewhat understated in the remarkable impact that this group of GS events demonstrates. The win percentage rate of figure 1a is really very promising, and the impact of GS overexpression in the four most widely studied transgenic events was nearly 4%. This represents 2 to 3 years of breeding progress for grain yield. It seems plausible from the data presented that a prioritized GS1 overexpressor could be identified which would consistently provide yield advantage in a majority of environments with minimal potential risk of lower yield. I believe that this level of improvement in performance would be appreciated by farmers as varietal replacement of hybrids in the market place is routinely occurring based on smaller incremental improvements. I do appreciate that the authors acknowledge the existence and importance of the trait x environment interaction they report, but I feel that they perhaps overstate the implication since this exists when comparing any two entities in any wide-scale field trial evaluations.

We have limited this overstatement by modifying the Discussion on page 16, 17 and 19.

The primary limitation of the association mapping study is that the experiment was only conducted in two locations. This is very limited for the

question that the researchers are trying to address. That said, the MAGIC population was well designed and the testcross approach appropriate. The finding that both Gln1.3 and Gln1.4 native alleles associated strongly with yield in one of two locations contributes to the evidence base that GS1 variation is important. Although this would have been strengthened by additional environments, the authors conclusions are appropriately made. The results suggest an important role of GS1. Taken together with the transgenic results, this does point to a plausible target for further validation work for either a marker-assisted breeding or gene editing approach.

We have strengthened the finding by adding three new environmental conditions corresponding to three different locations over three years of experimentation as summarized in Table 1. The text has been modified accordingly in the Results section page 13 and 14, in the discussion on page 18 and on the abstract.

I do think that the results of this study will influence thinking in the field. There is general skepticism about the potential for manipulation of specific genes to increase grain yield, and documenting a ~4% yield improvement by overexpression of one gene is an important piece of evidence for those working in this space to consider.

I do feel that the statistical approaches used are appropriate. I also appreciate the level of detail provided by the authors given the multi-year, large scale effort of this project. Although it would be very difficult to exactly replicate the work, I feel that it is described in adequate detail to give knowledgeable practioners a very good idea of how the authors carried out the research.

Respectfully, Michael (Mike) Olsen, CIMMYT Global Maize Program